# The Value of Traditional Fishing from the Perspective of Cultural Heritage: A Case Study of Seine Fishing in Ichinomiya-cho, Kujukuri-hama

**Lin Bai ***, Yirui Han and Konomi Ikebe

Graduate School of Horticulture, Chiba University, Chiba 271-8510, Japan
* Correspondence: caca0941@chiba-u.jp

**Abstract:** In this study, the significance of the preservation of traditional fishing is investigated in depth. Further, the practical value of traditional fishing and its intrinsic and irreplaceable nature is examined. This was achieved by utilizing the practices of seine fishing that take place at Ichinomiya-cho, Kujukuri-hama, as a case study and also by clearly showing how the economic value of traditional fishing is derived and its fundamental nature. In addition, traditional fishing has a non-economic value in terms of nature conservation, local social maintenance, human relationship enhancement and local landscape formation. As such, the irreplaceable nature of traditional fishing in terms of work, culture and landscape is clarified in this study.

**Keywords:** traditional fishing; cultural heritage; irreplaceability; multiple values; cultural landscape





## 1. Introduction

Around 70% of the Earth's surface is covered by water; additionally, the oceans hold about 96.5% of all Earth's water. We are all connected by the ocean. We depend on it for food, recreation and the maintenance of many diverse livelihoods. Fishing and hunting are more primitive productive activities when compared to farming. Piles of shellfish, fish bones and other debris left behind by ancestors at ancient human coastal dwelling sites are known as "shell mounds". The findings suggest that our early human ancestors used rafts, or some type of boat, in order to conduct offshore fishing activities. According to the white papers of FAO, fish and shellfish account for 17% of the animal protein consumed by humans [1,2].

In addition to providing food, the growth of fishing culture has left behind a rich cultural history. It has been recognized that the development of fishing and fisheries, as well as the activities of seafarers and fishermen, have colored in many ways and contributed to its diverse political, economic and social developments [3,4]. As such, activities related to the conservation and utilization of traditional fishing are becoming a more noticeable and increasing trend due to the promotion of legislation on cultural heritage conservation and due to the goal of revitalizing farming and fishing communities. Maritime and coastal cultural heritage, encompassing land and sea and underwater, is a particularly important part of our cultural resources and requires proper valorization in order to play its role in sustainable development for poverty reduction, livelihood promotion, education and environmental protection and helps to promote people's sense of identity and place attachment [3,5].

Regarding cultural heritage, there are 27 intangible cultural items indexed by the keyword fish in the World Heritage Database, comprising two fishing activities, six fishing-related activities including shipbuilding, five fish-related culinary culture and 13 fishing-related rituals and folk culture [6]. These fish-related World Heritage sites appear to classify fish-related culture into various subcategories, including fishing, boat building, cuisine and folklore. To collect aquatic items, however, fishing as a human activity has a number of

components, including objects, actions, fishing gear, fishing techniques and fishing boats [4]. The foundation of traditional fishing should receive more attention when discussing these intangible cultural heritages (ICH).

In this regard, the introduction of cultural landscapes within the legal instrument of the 1992 UNESCO World Heritage Convention is widely hailed as a landmark achievement. Cultural landscapes are a new type of world heritage to reveal and sustain the great diversity of the interactions between humans and their environment, to protect living traditional cultures and preserve the traces of those that have disappeared [7]. They are "the combined works of nature and of man" as defined in Article 1 of the Convention. As of 2023, 121 properties with six transboundary properties (one delisted property) on the World Heritage List have been included as cultural landscapes [7]. A total of 10 traditional fishing areas are included in the cultural landscapes inscribed on the World Heritage List. As human behavior interacting with the environment in an organically evolving landscape, traditional fishing fits the concept of organically evolving landscape in the second type of cultural landscape. The BudjBim cultural landscape, for example, contains elements of the Gunditjmara people using the rich local volcanic rock to build channels, weirs and dams to manage water flow in order to systematically capture, store and harvest Kooyang [8]. This suggests a new direction to consider traditional fishing as a cultural landscape to study the interplay of traditional fishing behavior on the land, regional society and human behavior.

In addition, in 2002, the Food and Agriculture Organization of the United Nations (FAO) launched the Globally Important Agricultural Heritage Systems (GIAHS) Conservation Project, which aims to establish globally important agricultural cultural heritage sites and their associated landscapes, biodiversities, knowledge and cultural conservation systems. Further, the aim is to recognize and protect these practices worldwide, as the basis for better and more sustainable management. As of May 2022, 65 globally important agricultural cultural heritage sites have been recognized by the FAO. There are only seven GIAHS including traditional fisheries, concentrated in Japan and China. The focus is on systems, ways and means of human influence on nature and coexistence. A GIAHS is a living, evolving system of human communities in an intricate relationship with their territory, cultural or agricultural landscape or biophysical and wider social environment [9].

However, relatively few traditional fishing sites have been designated as UNESCO World Heritage Sites and many more are in danger of disappearing. The issue of traditional fishing's low productivity has grown worse as a result of advancements in fishing technology, such as powered fishing boats and the modernization of fishing gear. If we want to keep the regional traditional fishing practices alive after they were abandoned as business fishing, proving that it still has other values, such as economic, social and cultural ones, should be the first step. According to Celeste et al., fishery societies have changed and diversified their economy due to a lack of profitability. Tourism is an alternative to reactivate fishing from its reinterpretation as cultural heritage [10]. Moreover, traditional fishery can be used as a tourism resource and become a fishery heritage to promote the sustainable development of fishery, better help people understand and protect the cultural value of fishery and contribute to the identity of fishermen [11].

However, this does not demonstrate that every traditional fishing activity has a similar value. To prove the worth of each unique case as a piece of cultural heritage, we require a method of value judgment that can be applied to particular traditional fishing practices.

We review the literature on how to measure the value of intangible cultural heritage (ICH). Masoud and Xiao investigated the economic value brought by ICH as cultural capital [12,13] and Heredia-Carroza designs an empirical methodology to measure the perceived value of ICH [14]. Lenzerini noted that understanding the intrinsic meaning and value of ICH requires an understanding of human factors such as self-identification, the history and social evolution of the group corresponding to the heritage and the close connection of the heritage with the unique identity of its creators and holders [15].

Measuring the value of ICH involves not only the economic benefits it generates but also the exploration of its significance to local people and society. This is consistent

with the GIAHS and the idea of a cultural landscape. Cultural landscapes depict a deep and enduring bond between humans and their natural surroundings [5]. Combining GIAHS theories, the three primary values of traditional fishing are the most crucial thing to study—(i) local and traditional knowledge systems; (ii) cultures, value systems and social organizations; and(iii)landscapes and seascape features [4].

When it comes to traditional fishing, related cultures, value systems and social organizations are discussed in the context of folklore, climatology and ecology. The study on ancient Chinese fisheries illustrated that ancient ecological and ethical knowledge can be found in the conventional fishing approach, which has been practiced for millennia [16]. Local fishing communities' resilience to potential climate change is further boosted by the experience that generations of fishermen have amassed [17]. Besides, traditional fishing practices and associated folk activities like sea sacrificial ceremonies also strengthen the unity and effort among fishermen and generate a cohesive force from their hearts to build confidence to defeat any terrifying waves and storms [18]. Sidney noted that traditional fishponds have been serving not only as a source of traditional local food but also adding to the conservation value at large [19]. Mulazzani believed that fishermen are often considered to be the guardians of coastal customs, traditions and an age-old way of life. Bennett noted that elements linked to cultural heritage may include fishing harbors, boats, fishermen themselves and the related buildings, festivals, traditions, traditional trade and barter [20].

These studies on the importance and measurement of the value of traditional fishing have connections to regional society, folklore, ancient ecological ethics and ecological conservation. However, no research has examined traditional fishing as a whole; each research case examines a particular aspect of traditional fishing.

Up to this point, no significant work has been undertaken in the field of assessment of any traditional fishery's total value as intangible cultural heritage and the necessity of preserving it. Besides, none of the above-mentioned approaches takes into consideration the viewpoint of cultural landscapes.

Within the context of the above background, the significance of this paper is that it discusses the values of traditional fishing as a human activity and ICH.The uniqueness of this study is in the method this article suggests for examining the significant worth and irreplaceability of traditional fishing from the perspective of cultural heritage, particularly cultural landscapes that traditional fishing was a production practice that affects local culture, society and landscape. Additionally, the method for measuring values that was created in this study for traditional fishing might also be used to estimate the value of traditional forestry, agriculture and animal husbandry asareference.

## 2. Materials and Methods

We begin by exploring the concept of traditional fishing and the specifics of its value. The application of the concept and value judgments is then used to comprehend how traditional fishing affects the local economy, society and environment as well as how irreplaceable it is as a cultural heritage, illuminating the necessity for a thorough investigation and preservation of traditional fishing. Seine Fishing in Ichinomiya-cho, Kujukuri beach, a centuries-old Japanese traditional fishery, is used as a case study to demonstrate this.

### 2.1. Related Concepts and Concept

### 2.1.1. What Is Traditional Fishing (伝統漁撈)?

Traditional fishing (伝統漁撈), as conceptually utilized in this study, is defined as "fishing that is characteristic of the area or that has had a significant impact on the productive life of the local people for a long time by means of archaic traditional fishing and old fishing methods".

### 2.1.2. Concept of the Value of Traditional Fishing

The essence of traditional fishing is the production work of fisheries, which is an economic source of income, as well as a source of direct and indirect fishing revenue. Shikita advises that attention should be directed to functions and workings other than the production function of the Japanese fishing industry [21]. The value of social, psychological, historical and cultural aspects, such as faith, art, living in harmony with nature and environmental logic are all important aspects for human beings that need to be considered.

Therefore, in this study, three types of economic value are examined. These are the direct monetary value from traditional fishing operations; indirect monetary value from the distribution, processing and branding of fish catches; and the value gained from fishing experiences and related fishing stays that could become a tourism industry.

When regarding the non-economic value, the specific functions and roles of traditional fishing were clarified by the multifaceted functions of the fishing industry and fishing villages. Fisheries and fishing villages have long played the role of providing a variety of multifaceted functions to people, such as border surveillance and maritime rescue in order to protect lives and property; providing places for health, recreation, exchange and education; and providing a stable supply of safe and fresh marine products to the people [22].

In this study, it will be demonstrated through specific cases how traditional fishing has these non-economic aspects.

### 2.1.3. Irreplaceability

However, possessing multiple values does not mean that there is a need for preservation either. We need to prove the necessity of keeping it as traditional fishing practices in the present day by discussing the irreplaceable parts of traditional fishing. Irreplaceable, in this context, means that in losing it, there would be great regret and loss. Apart from its economic and utilitarian value, it is necessary to clarify what kind of irreplaceable being traditional fishing possesses.

Heidegger distinguishes between non-essential and essential modes of being [23]. Ryo Oda advised that the latter is a replaceable way of being created by the instrumental ways of modern society, while the former refers to the irreplaceable individual [24].

In addition, when it comes to the "irreplaceability of the individual" there are two types of individuality: "comparable" and "incomparable". The former is positioned as "particularity" and the latter as "singleness" [25,26].

By definition, the essence of traditional fishing is found in the working activity of fishing. This activity is what supports the livelihood of local fishermen and is deeply related to various human lives and folk culture, including local food culture and folk events [27,28].

As such, when it comes to traditional fishing, first and foremost it must be understood as a livelihood-based activity. Due to this, the irreplaceability of its individuality requires further examination.

In addition, from the conceptual perspective of the cultural landscape, the nature of traditional fishing involves how it affects the local environmental patterns and landscape formation. As such, examining the nature of the landscape in the area is part of this study. "Something individually special or unique in the fundamental way of fishing that reflects the environmental form and shape (landscape), the history and culture of fishing through traditional fishing" is how the irreplaceability of traditional fishing is defined.

In this study, we have examined the irreplaceable nature of traditional fishing, mainly in terms of its production work, as well as its historical and cultural aspects in regard to traditional fishing and landscapes.

The analysis process of irreplaceability of traditional fishing in this study has showed in Figure 1.

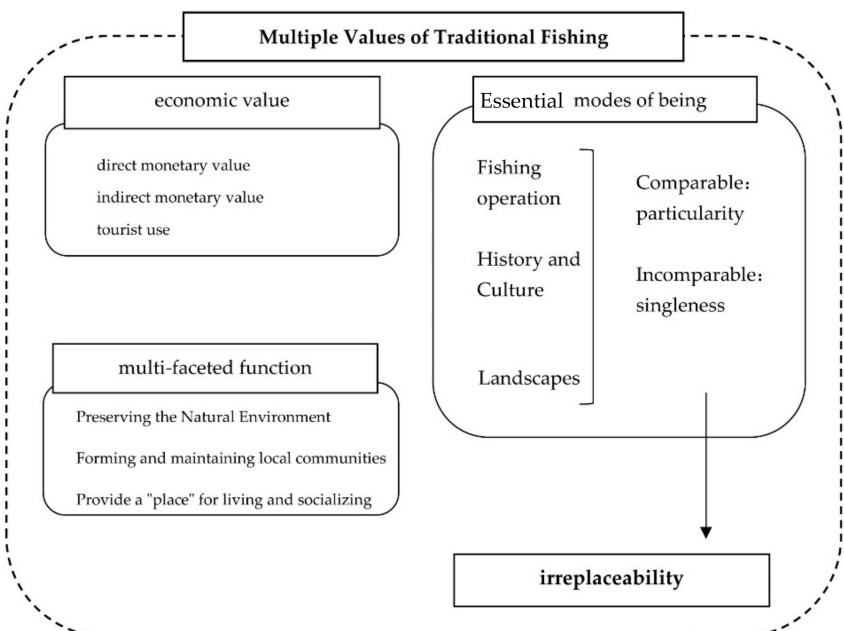

**Figure 1.** Analysis process of irreplaceability of traditional fishing.

*2.2. Research Subjects and Information Collection*

2.2.1. Target Selection

Japan shares the same issue as the rest of the world in that traditional fisheries are still not widely acknowledged, despite the fact that important traditional fishing landscape sites have been chosen as important cultural landscapes and that traditional fishing gear and technologies have been recognized as national cultural assets [27]. Japanese GIAHS have been emphasizing the landscape aspect [29].

As it is surrounded by the sea, Japan has formed coastal settlements, developed fishing and navigation technology and culture and developed fishing methods and traditional fishing culture that are rare in the world. Among FAO, the Nagara River Basin Fisheries System and Lake Biwa System are systems with fisheries and agriculture that are nurtured by forests, villages and lakes. They all show a complex and diverse fishing culture that is closely related to the life and production of local people [30]. At the same time, people are starting to direct more attention to this ancient fishing method. In total, 15 important cultural landscapes related to traditional fishing were included in the "Nationally Selected Important Cultural Landscapes" [31].

In addition, Kujukuri Beach in the northeast Chiba Prefecture is one of Japan's representative sandy beaches (as per the Institute for Ocean Policy Research, Ship and Ocean Foundation, 2005). Through seine fishing, a fishing culture has traditionally grown here; however, today, this traditional fishing activity only exists as a tourist activity that can be engaged in and is held once or twice a year. Seine fishing's plight is illustrative of the struggle of traditional fishing to survive in the future since it is currently only supported by a few annual events. There is a chance that other more alluring tourism events will take their place. The technology may also be lost as a source of tourism if the fishermen who mastered it become elderly and no one is left to carry on the tradition.

At the end of the Middle Ages, seine fishing techniques were introduced and the Kujukuri area was developed as a beach and seine fishing ground. From the early Edo period to the 20th year of the Meiji period (1868–1912), the beach seine fishery on Kujukuri Beach experienced several difficulties in terms of good and bad catches. It was at its peak during the Tempo period, when it reached the height of its get-rich-quick prosperity. Due to overfishing, institutional issues and technological delays, the beach seine fishery gave up its leading role to the beach seine fishery in the mid-to-late Meiji period; this area is no longer as active for fishing as it once was. These days, seine fishing is practiced as a

recreational activity, whereby there is a recreation of the production activities of the past. In addition, although seine fishing is a cultural asset designated within Ichinomiya Town, it has not been designated or surveyed as a cultural landscape. At present, as with many traditional fisheries, there are few studies or plans for their preservation.

In addition to the above, we also conducted several field surveys. Moreover, a wealth of information was obtained through the cooperation of the local seine fishery preservation society, as shown in Table 1.

**Table 1.** Books and material from regional organizations.

| Name of the Data | Publisher | Publishing Place | Date of Publish |
| --- | --- | --- | --- |
| Furusato Now and Then | Hidemi Hasegawa | Joso Ichinomiya Local Studies Group | April 1988 |
| Report on the Complete Survey of Buddhist Sculptures | | | |
| Cultural Assets of Ichinomiya Town | | | |
| Searching for Disasters in Kujukuri | | | |
| Local Characteristics of the Facility Horticulture Community in Ichinomiya Town, Chiba Prefecture | | | |
| A Catalogue of the Kano Family Archives | | | |
| Bird's-eye View of Ichinomiya Town | | | |
| Ichinomiya-cho Small Character List | | | |
| Chiba Prefecture Ichinomiya-cho Bird's-eye View Showa | | | |
| Map of Ichigunmiya-cho, Chiba Prefecture | | | |
| Home town | Hidemi Hasegawa | Joso Ichinomiya Local Studies Group | December 1981 |
| One's old home | Akira Kanada | Joso Ichinomiya Local Studies Group | October 2004 |
| Ichinomiya Town History | | | |
| Society and Culture of East Kamisou | | | |
| Map of Ichinomiya, Nagasun-gun, Chiba Prefecture | | | |
| Ichinomiya-cho Old Map 1 | | | |
| Ichinomiya Town Old Map 2 | | | |
| Ichinomiya Hongo Village Old House List | | | |
| Copy of the monument | | | |
| 15 old photos of Ichinomiya-cho | | | |
| 18 old photos of Ichinomiya Town | | | |
| 4D Maps Ichinomiya 2014 | | local cross | June 2014 |
| Looking back on the history of the city in the Meiji, Taisho and Showa eras | | | |
| Survey Report on Historical Buildings in Ichinomiya Town H25 | Higashi-kamiso Cultural Heritage Comprehensive Revitalization Project Committee | | 26 March 2014 |
| Illustrated History of Chousei and Isumi | MIURA Shigekazu, KATO Tokio | local publisher | 24 February 2010 |
| 100 Years of Mobara, Katsuura, Chousei, and Isumi | Shigekazu Miura | local publisher | 31 May 2002 |
| Map of Cultural Assets of Ichinomiya Town | Ichinomiya Town Board of Education | | March 2014 |
| Map of Cultural Assets of Ichinomiya Town | Ichinomiya Town Board of Education | | March 1999 |
| Map of Ichinomiya Township and Merchant Villages in the First Year of Meiji Era | | | |
| Map of Old House Registry of Ichinomiya Hongo Village in the First Year of the Shoho Era | | | |

2.2.2. Information Collection

Additionally, in this study, we collected related information regarding beach seine fishing from documents in the collection of the National Diet Library, documents in the collection of the history in Ichinomiya Museum of History and Culture and documents available on the official websites of the Ichinomiya Town Office and the local fishery cooperative.

## 3. Results

### *3.1. Beach Seine Fishing*

#### 3.1.1. Fishing Method

Beach seine fishing is a method of catching fish by pulling nets from the sea while remaining on land. As shown in Figure 2. The boats leave the beach with big nets in order to build a gap between the five nets. Dozens, or even hundreds, of people waiting on the land pull a net toward the land, thereby driving a school of fish into the net and thus catching these fish [32,33].

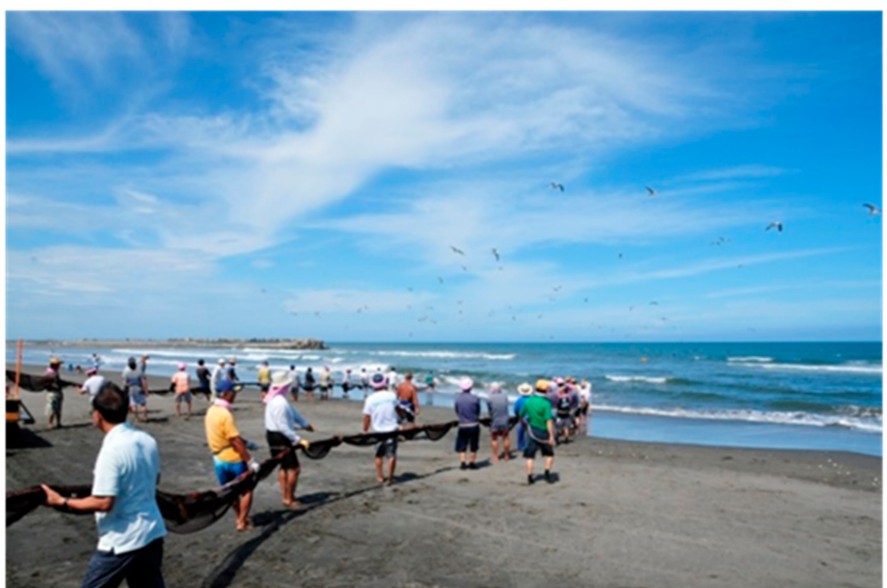

**Figure 2.** Beach seine fishing at Ichinomiya (Source: Ichinomiya Town Hall official website).

#### 3.1.2. History of Seine Netting at Ichinomiya

The practice of beach seine fishing was developed in Ichinomiya over hundreds of years. It has gone through periods of great prosperity, as well as several periods of decline when production became low. Table 2 shows the development of the local fishing industry that was made based on historic documents of Ichinomiya Town [34].

#### 3.1.3. Fishing Landscape of the Past

According to a seine fishing Ema (Tamamae Shrine, Ohara-cho Collection), during the period of seine net development, seine net fishing activities were thriving with the cooperation of several hundred fishermen. In addition, each boat was worth a thousand gold pieces. The boat was driven and the nets were deployed according to the fish population; furthermore, the people worked on the beach pulling the nets. From the early Edo period (1600s) to around the Meiji 20s—i.e., (1880s), a period of about 300 years—the working landscape of beach seine fishing could be seen on Ichinomiya Beach, as shown in Figure 3 [33].

**Table 2.** Beach Seine Fishing History Chronology.

| Period | Years | Development of the Seine Fishery |
|---|---|---|
| Early period (of development) | 1555–1558 | A Kansai fisherman named HisasukeNishimiya drifted ashore at KujukuriUra and introduced seine fishing. |
| | 1661–ca. 1699 (Kanbun-Genroku period) | With the height of prosperity |
| | 1703 (during the year of Genroku) | The five great powerboats were built |
| Prosperous period | 1603–ca. 1710 (End of the middle century, early Edo period) | Developed as a sandy beach seine fishing ground |
| | ca. 1710–1790 (mid-Edo period) | Fishermen's Barn Settlement Formation |
| | 1615–1624 (Empo and Tenwa period) | Increase in the number of nets |
| | 1716–1741 (Kyoho and Genbun period) | Great catches were seen |
| | ca. 1670–1730 (Late Horeki-Meiwa, An'ei period) | Poor catch |
| | 1790 (the second year of the Kansei Era) | Various protection measures were implemented to help seine fishermen recover |
| | 1804–30 (Bunka, Bunsei period) | Prosperous period |
| Decline | 1887 | Gradual decline |
| Preservation | 1982 (Showa 57) | Seine Fishing Preservation Association formed |
| | 30 March 2012 | Designated as a town cultural property |
| | Present | Conducted as a tourist seine fishery |

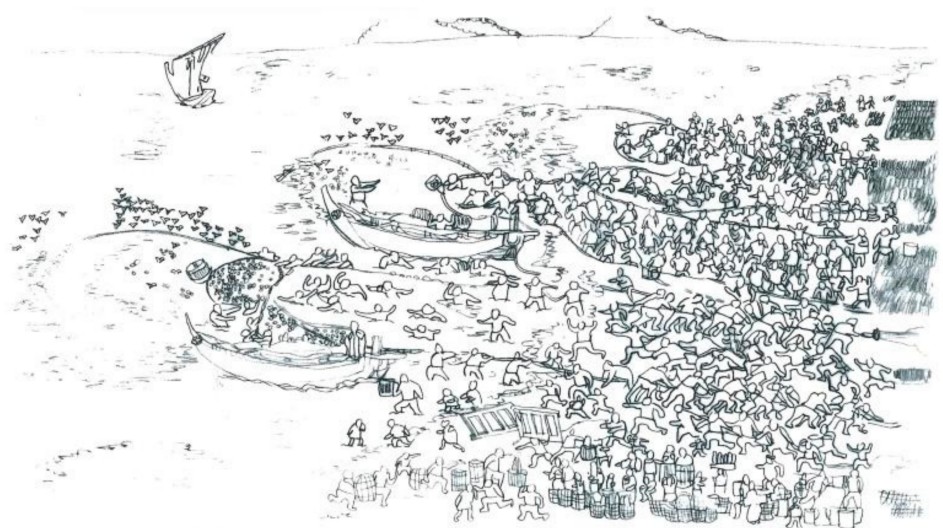

**Figure 3.** Beach seine fishing in the olden days.

*3.2. Pluralistic Value*

3.2.1. Monetary Value of the Economy

(a)   Capturing direct monetary value from traditional fishing operations

The history of beach seine fishing shows that it was a major local fishery for more than 200 years from the Edo period as a high-catch, high-profit livelihood. In the An'ei era (ca. 1770–1780), there were 440–880 beach seine fishermen. Further, in 1827, the number of beach seine fishermen reached approximately 1760 on the Ichinomiya coast. In the heyday of the Tempo era, beach seine fishing was a thriving business that made people rich in a day's work [33,34].

However, beach seine fishing gradually declined after around 1887. According to records from 1933, there were 12 beach seine fishing boats [34]. Further, by 2005 there were zero. According to the Ichinomiya Town Comprehensive Plan, the current situation is that the town's main fisheries industry is tourist beach seine netting, as well as the collecting of seed eels at the mouth of the river [35].

Today, beach seine fisheries do not have a direct monetary value from the catch.

(b)  Capturing indirect monetary value from traditional fishing (income captured from the distribution, processing and branding of the sales of catches)

There were commodity exchange flourishes and emphasis from the Shogunate on economic policies during the period of peace. In Ichinomiya, commodity self-sufficiency was afforded; further, the Tamamae Shrine Monzenmachi, as a regular market, was an important passageway in the Boso district.  For these reasons, at the beginning of the 17th century, a regular market called 'Goju-ichi (五十市)' was established to sell fish, marine products from Kujukuri Beach and agricultural products from the region, as well as general merchandise [33]. This was an important place for the disposal of beach seine catches.

Due to the development of the fishing industry and the increase in marine trade, the size of the regular market became larger. A commercial area was gradually formed in the area and several fishing shops were built.

Beach seine catches have inevitably been part of the history of geographical development, as they are processed in the immediate area and transported to more distant places. Diverse cultures have also developed due to the impact on the lives of the fishermen and those involved. Now that the fishing industry has converted to tourism, related industries, such as sardine processing and distribution, are no longer what they used to be. However, the distribution roads and shopping streets that were developed during the prosperous period have, over the years, become important streets in the Ichinomiya area. In this respect, the practice of beach seine fishery today has contributed to the development of the region, although it has lost some indirect financial benefits.

(c)  Tourism use

Two or three seine fishing trips are currently conducted in Ichinomiya Town each year on the Shinhama Coast (south of the Ichinomiya Coast) as a leisure activity (though events may be canceled due to typhoons or COVID-19). Visitors can experience the production activities from the past; further, anyone can participate, free of charge.

When it comes to economic earnings, Ichinomiya has 96 establishments (478 employees) [36] in the accommodation industry—with a mix of different cultural spheres—including coastal summer resorts, farmers and fishermen. In order to promote tourism in the town [35], the free experience of seine fishing is one of the initiatives in place to attract tourists from inside and outside the prefecture.  As such, many people visit the sea of Ichinomiya during its fishing season in order to enjoy seine fishing [37].

Therefore, although the seine fishing experience is free of charge, it serves to attract tourists and promote tourism to the coast, thereby clearly bringing economic benefits to the region.

3.2.2. Multifaceted Functions of Traditional Fishing

Ishihara pointed out that fisheries and fishing villages have multifaceted functions [27]. As such, we clarified the specific roles of traditional fishing through the following four functions.

(a)  Providing fish, which is the original function of the fishing industry

The current beach seine fishery is regarded as a tourist fishery and the fish caught are distributed free of charge to the participants. In addition, the catch is so large from one net that it provides extra income for dozens of dual-income fishermen, as well as providing ingredients for home cooking.

It can, therefore, be said that traditional fishing has the function of providing fish (i.e., marine products).

(b)  Conserving the natural environment

In terms of fishing techniques, beach seine fishing is not an environmentally friendly fishing method, due to the fact that it waits for sardines to migrate to the coast. However,

the current beach seine fishery is restricted to a few trips per year and the catch is very limited. In addition, according to the Ichinomiya Coastal Management Ordinance [38], permission to fish is only granted by the town mayor if the fishing does not damage the coast and surrounding environment. Due to the above, there is no environmental or resource damage caused by beach seine fishing.

(c)　　Formation and maintenance of the local village

(c-i) Formation and development of the fishing village in the context of traditional fishing

The size of Ichinomiya's 'Goto-ichi' (Periodic Market) gradually expanded with the prosperity of seine nets in the 17th century, despite the abundance of agricultural, forestry and marine products. In the mid-Edo period, seine fishermen formed barn settlements in narrow, long areas close to the beach, where they would stay during the fishing season [33]. The locations of Goto-ichi and Naya Village (the formation of barn settlements for fishermen 納屋集落) overlap at the Ichinomiya Tamamae Shrine gate street. Further, the seine fishing industry and Gojuichi are mutually beneficial. In addition, the wealth of Ichinomiya has gathered and developed along the street, including business shops, sake shops and pawn shops [33,34]. These landscapes are related not only in terms of location but also in terms of human activity and wealth reserve flow.

The beach seine fishing industry in Ichinomiya has long been one of the main livelihoods supporting the lives of the district's inhabitants. It is clear that it played an important role in the formation and development of the district's settlements.

(c-ii) Maintenance of fishing communities in traditional fishing

Ando advised that, unlike ordinary organizations, human relations in local-type production organizations are neighborly relations that have been formed historically and are deeply connected to social and economic relations in the local area [39].

The relationship between fishermen has transformed from the old lordship and net-owner-subordinate relationship to the modern model of fishing associations under communal fishing rights. Furthermore, the relationship between fishermen has been transformed in their livelihood. This has led to the formation of the present-day ties within fishing groups and the maintenance of the present-day relationships within fishing communities based on the sharing of economic earnings. With regard to local fishing groups, before the war, coastal fisheries were mainly family-run businesses. Further, to this day, private businesses still account for more than 90% of the total, which is still high even though the proportion of family-run businesses has decreased [40]. Kase advised that this is due to the strong full-time character of fishing households in contrast to agriculture, especially with regard to the proprietors themselves [41]. This reflects the strength of blood and geographical relations in the organization of seafarers in the particularities of fishing labor, as well as the local character based on the history of the formation of the fishermen as a geographical fishing group [42].

It also shows that traditional fishermen groups have strong intra-group ties within kinship-type groups due to their more ordinary employment relationships. Traditional fishing is mainly inherited by small kinship groups, which are considered to be highly cohesive to the local community due to the depth of kinship ties.

(c-iii) In terms of the spiritual aspect, the maintenance of local social relationships is achieved through folk culture

In Japan, fishery products are closely connected to daily life and the country has a rich water bounty, thereby making it a world-class fish-eating culture.

The Kujukuri Beach large beach seine fishery was operated from 1555–ca. 1558. The sardines caught were not only edible fresh as Sashimi but were also processed into dried sardines for the purposes of fertilizers and fish meal, which were shipped nationwide, thereby providing a valuable source of income for the local people [43]. The factors that shape food culture are broadly classified into three groups: "natural conditions", "human technology" [44] and "social conventions". Moreover, the beach seine fishing at Ichinomiya is thought to have played a role in the formation of the local sardine food culture.

The food culture was inherited and the relationship between generations was good through mutual exchange [45]. While some food culture was formed in relation to commercialized foodstuffs on the basis of the society as a whole, it was also observed that it was established on the basis of traditional areas, such as neighborhoods, villages and towns [46]. It is thought that the sharing of food culture is related to the multiple seine fishing, due to the fact that a source of local pride plays an important role in maintaining relationships in households, neighborhoods and communities.

Additionally, Tsurigasaki Beach is said to be the place where Tamayorihime (玉依姫), the deity of Tamasaki Shrine (and the first shrine in Joso Province), landed. Furthermore, there are legends of her bestowing good fortune at sea and in the mountains [34]. The beliefs in Tama Omi, the goddess of the sea, and prayers for peace and tranquility on the sea have been passed down to this day. The activities of shrines, such as theTamasaki Shrine and Namiya Shrine that are related to fishing show the legacy of the fishing boom period along the Kujukuri coast.

In the mid-Edo period, as the seine fishing industry began to flourish, the production of farmers and fishermen increased. Further, the net owners, who had become wealthy, were in awe of the literary figures and treated them with respect. The area became a busy holiday resort, known as "Oiso in the East" and was visited by prominent politicians, military officers, businessmen, scholars and artists. Leading literary figures of the Meiji era, such as Ryunosuke Akutagawa, Sa-chio Ito and Kotaro Takamura, visited the area. During his stay, Akutagawa often sent long bulbous letters from the villa to Tsukamoto—who later became his wife. His memories of Ichinomiya appear in his works, such as Smile, On the Edge of the Sea, Genkaku Sambo and Mirage [47].

Seine fishing for sardines flourished along the Kujukuri coast from the Edo period to the beginning of the Showa period. Labor songs and big-fishing celebration songs were sung during this period. The lyrics of the songs include references to scenes of the sea and fishing, as well as the scenery of Higashinamiami [37]. The Big-Fishing Celebration Song was designated as a prefectural intangible folk cultural asset back in 1965.

The Ichinomiya Ondo (一宮音頭) was written by Shogo Shiratori, a poet who was evacuated to Higashinamiami [37]. The lyrics of this song, composed around 1949, mainly by the women's association, mention the sea, festivals and old events in Ichinomiya.

From the above, it can be seen that local seine fishing played an important role in influencing literature andart and enriching the spiritual life of the local people.

(d) Providing a variety of "place" for livelihood and exchange

Ikeya defined subsistence activities as "activities in which humans acquire food or produce or exchange goods essential for survival" [48].

With reference to Nobuhiro's discussion [46], in this study, the various activities of traditional fishing are categorized as follows:

- Fishing operation activities: fishing capture activities
- Sub-system 1: the preservation, processing and treatment of catches; distribution and circulation; consumption; and disposal
- Sub-system 2: fishermanhouse-building activities; fishing boat and fishing gear-making activities; exploration activities; skill transmission activities; ritual activities; play and leisure activities; and sleep and rest activities.

With these activities, traditional fisheries provide a variety of "places" for diverse livelihoods and interactions.

It was also advised by Mitsuhashi that fishing is usually a recreational productive activity. It is a distraction, with a strong playful character, usually participated in as a hobby undertaken only by a small group of people [49].

Focusing on beach seine fishing, fishing operations are conducted on the sandy beaches and coastal waters of the Ichinomiya coast as fishing grounds. This is in addition to it providing opportunities for technological transmission and intra-group exchange. Moreover, the natural background and the fishing landscape of the fishing grounds provide

opportunities for the purposes of recreational fishing, as well as fishing experiences as a local tourism resource. Along the Kujukuri coast, there are traces of the barn settlements (納屋) created by seine fishing in the Edo period [50]. The places of play and leisure for the old-time fishermen, such as folkways placesand old shops, as well as the places for festivals and Shinto rituals, are all still often the places where village life and village events take place.

From the above, it is clear that traditional fishing has three multifaceted functions:

1. To conserve nature from the perspective of resource response and environmental protection;
2. To form urban areas and maintain society from the perspective of the formation of fishing villages, and the maintenance of local human relations;
3. To provide a variety of 'place' for various forms of life and interaction.

### 3.3. The Original Nature of Beach Seine Fishing

3.3.1. Beach Seine Fishing Operation

In this study, we investigated and analyzed the elements of traditional fishing (i.e., fishing method, catch, fishing gear, fishing boat, fishermen, and fishing season) in order to explore its original nature.

(a) Fishing methods, gear and vessels

In regard to the relationship between fishermen and nature in the past, it can be seen that in their pursuit of a symbiotic way of life, they knownature and live in harmony with it.

They must use fishing gear under the guidance of the law regarding fishing and they must also fish away from the shore in a fishing boat. The fishing techniques and methods are irreplaceable.

In order to increase the strength of the ropes, both the nets and the seines can now be made from modern materials. Therefore, the materials of fishing gear and fishing boats are replaceable.

(b) Fishermen

When we speak of fishermen, we do not refer to an individual fisherman, but to a group or collective of fishermen.

Seine fishing is a collective effort and human cooperation is the most important part of a successful fishery. The point of successful ground trawling is that the fishermen determine the location of the sardine schools based on the currents and environment. Further, the people pulling the nets on the beach work together in order to properly complete the pulling. In this regard, fishermen's ingenuity and collaborative skills are demonstrated.

(c) Catch

The quality of traditional fishery catchestendto be better and fresher due to careful handling and less damage to the catch.

Ichinomiya Town has flourished in seine sardine fishing since the Edo period and the "sardine culture" is still alive. The current situations that seine fishing normally takes place in is the morning for about two hours. The participants are provided with the caught fish as souvenirs, free of charge. The fresh fish caught in the morning looks delicious to eat raw on the spot or to take home and cook. The fact that it is free of charge may also increase interest in sardines as a food source.

(d) Fishing season

The fishermen were organized according to their experience, as handed down from generation to generation. Through this, the habits of sardines and the best time to fish were determined. Additionally, their fishing calendar can also prevent the shortage of resources due to overfishing. At a time when society as a whole was not conscious of sustainable development, the fishermen of the traditional fishing industry were conscious of respecting nature and protecting fishery resources.

(e) Fishing grounds

In terms of environmental conditions, traditional fisheries have responded to a variety of fishing methods in different environments and have thus formed diverse fishing grounds.

The water environment has created a good fishing ground for sardines in the offshore area of Ichinomiya.

The beautiful sandy beach is flat and free from obstacles as a base on land, which is ideal for a large number of people to gather. In addition, the fishermen's groups can smoothly pull nets on land even if there are more than 100 people.

The adaptation and use of the environment by traditional fisheries arevery much in tune with the specific nature of the environment itself and the habits of the fish in situ.

3.3.2. On the Cultural Front

The culture of traditional fishing is difficult to define precisely. According to Kojien (Japanese Dictionary, published by Iwanami, 6th edition), culture is "the material and spiritual achievements that human beings have formed by modifying nature. It includes food, clothing, shelter, science, technology, learning, art, morality, religion, politics, and other forms and contents of life formation."

In the research field of fishing culture, the sea is regarded as the place of production, whereas the land is regarded as the place of life [51]. In other words, the culture of traditional fishing includes not only the contents of traditional fishing, but also the mode and contents of life that affect the employees and local residents, such as food, clothing and shelter.

According to "Culture and wetlands: a Ramsar guidance document [52]", wetland culture is classified into four types. Traditional culture is classified according to "Cultural Historical Significance and Cultural Inheritance Function of Fishery and Fishing Village [27]", which includes the description, folk knowledge, food culture, fishermen's beliefs, folk events and performing arts.

With reference to the above, this study considers traditional fishing to include four related types of diverse cultures:

- 1: Livelihood culture: including unique skills and knowledge as a means of earning a living;
- 2: Indirectly related livelihood culture: including food processing production, processing and its architecture, etc.;
- 3: Fishing village landscape: including the traditional living culture (food, clothing and housing), village and food culture, as well as daily activities;
- 4: Psychological outcomes related to traditional fishing: including folk customs, beliefs and social activities as well as fishermen's beliefs, folk customs and performing arts.

The culture of beach seine fishing is the core of the fishermen's livelihood, which includes knowledge of traditional fishing methods, techniques, tools and nature use. In the 330 years since the middle of the Edo period, many people in Ichinomiya have been employed in seine fishing and related businesses as a means of earning a living.

Beach seine fishing was designated as an intangible folk cultural assetof Ichinomiya Town on 30 March 2012. Further, its intangible cultural values, such as its fishing methods and techniques have been recognized.

As for indirectly related livelihoods, there were ancillary livelihoods, such as ship-building, net weaving and transportation and the processing of fish catches.

In the Edo period (1603–1867), the fishing of sardines usingseine nets began to flourish and people began to demand large quantities of sardines as fertilizer and food. Dried sardines, made from sardines, were used as fertilizer. Dried sardines were made by drying sardines in the sun on the beach. This type of dried sardines was created from the beginning of the Edo period, but it was not until the middle of the Edo period that they began to flourish. These dried sardines were sent not only to Edobut also to the Tohoku region and the Kansai region.

As for the traditional living culture, this section covers various aspects such as the form of settlements and towns, fishermen's residences, clothing and the food culture of traditional fishing.

As described in Section 3.2.2, the development of seine fishing led to the formation of barn settlements for fishermen. The locations of Goto-ichi (Periodic Market) and Naya Village overlap at the Ichinomiya Tamamae Shrine Monzen-gai. Moreover, commercial stores, liquor stores, pawnshops, etc., developed along the street. The seine fishing industry of Ichinomiya has long been one of the main sources of livelihood for its local residents. It has played an important role in the economic and cultural development of the area.

Regarding Culture 4, it is the local folk events, beliefs and social activities that are related to traditional fishing. With the development of the traditional fishing industry, the faith and traditional skills of fishermen with local characteristics were utilized. As shown in Section 3.2.2, many cultures related to faith and the literary arts began to pray for big catches; as such, safe voyages were left behind.

You can see the spirit of people who are in awe of nature and grateful for the fish they obtain from it.

In the traditional fishing industry, people have formed and inherited a view of nature and life that respects nature. Furthermore, the beliefs of fishermen in various regions have been developed along these lines also.

### 3.3.3. In Terms of Landscape

The traditional fishery landscape, in the narrow sense, belongs to the fishery landscape together with the fishing ground, fishing port and fishing village landscape. It is recognized as the scenery of working, fishing and casting nets in the fishing ground. Akimichi advised that, in the research field of fishing culture, the sea is the place of production and the land is the place of life [51]. Moreover, the actual fishing work behaviors are "linked in a chain and influence each other" in various places and these parts do not exist in isolation [53]. The landscape of traditional fishing work can be considered to include the landscape of the place of production and the landscape of the place of life in the "water area" and the "land area", respectively.

As a production activity that develops in the local water environment, seine fishing supports the lives of fishermen and is deeply related to various human lives and folk culture, such as local food culture and folk events. Its landscape is suitable for the definition of a cultural landscape, i.e., it is the cultural landscape of Ichinomiya Town.

Based on the relevant documents, we have sorted out the landscapes related to the traditional fishing landscape of Ichinomiya for the purposes of beach seine fishing, as shown in Table 3.

**Table 3.** Landscapes of Beach Seine Fishing in Ichinomiya Town.

| Landscape | Relation to Beach Seine Fishing | Related Historical Buildings |
|---|---|---|
| Beach seine fishing landscape | Seine fishing operation | |
| Goju-ichi | Catch sale and consumption | Street in front of Tamamae Shrine |
| View of Tamamae Shrine | Fishermen's religious activities, festivals, etc. | Tamamae Shrine |
| Naya Village | Fishermen's housing, tool storage | Naya |
| Retail stores | Nokantosei (concurrent occupation of Fishermen) | Well-established |
| Oiso villa area in the east | The men as head of fisherman's group were in awe of the literati and treated them well | Akutagawa Villa |
| Business district | Town development through exchange of fishing traffic | Commercial area in front of the gate of Tamamae Shrine |
| Ichinomiya-cho Seaside | Development and improvement of fishing beaches as surfing grounds and beach bathing areas | Beachfront bathhouse |

In addition to the fishing scenery, the practice of seine fishing left a variety of landscapes, such as the landscape of the Periodic Market of the old days, the landscape of the barn village of the fishermen, the landscape of the villa area named "Oiso of the east" on the coast and the landscape of the street with business stores, liquor shops, pawn shops, etc.

## 4. Discussion: The Irreplaceable Nature of Traditional Fishing

Firstly, when focusing on the beach seine fishing operation, we examined the irreplaceable elements with the elements of traditional fishing. The creativity and collaboration of traditional fishing cannot be replaced by automated technology, due to its technical complexity and the awareness of using nature through natural knowledge [54].

Human understanding and the appropriate use of nature—such as the acquisition and application of knowledge about fish habits in beach seine fishing and the use of the aquatic environment—are examples of the wisdom of labor through the accumulation and conclusion of the long-term experience of fishermen.

The careful fishing methods and group fishing that cannot be performed by individuals reflect the fishermen's human power. These creative acts, such as summarizing and passing on fishing methods gained from labor experience and careful attention to detail, cannot be replaced by modern techniques and tools.

In regard to the production activities of traditional fishing, its natural knowledge and human power are considered to be thus irreplaceable. Furthermore, these conventional pieces of wisdom are regarded as the pipeline to the future [55].

Secondly, the traditional fishing industry has formed and passed on a view of nature and life that respects nature. Moreover, it has developed the beliefs of fishermen in various regions. The people's reverence for nature and their gratitude for the fish obtained from nature can be seen. From the above diverse cultures, the practice of traditional fishing shows the "diversity of folk culture" as well as respect for nature. The cultural diversity of traditional fishing and the view of life that accompanies it are considered to be irreplaceable in terms of culture. Culture is considered to beat the root of all human decisions and actions and as an overarching concern in sustainable development considerations; it enables culture and sustainability to become mutually intertwined [56]. The most crucial foundation for ensuring the preservation of traditional fishing is culture.

In addition, as shown in Section 3.3.3, Ichinomiya Town has retained a variety of landscapes that are related to beach seine fishing.

Kameyama advised that the "LandSGhaft (風土)" refers to the character of the land and the individuality of the land created by the nature, social organization, culture and people's ways of life in a certain geographical space (region). Further, Kameyama further elaborated that the regions that still retain their lands truly show the original nature of human beings, which modern society has lost sight of [57].

It is clear that seine fishing is irreplaceable in its role in the formation of local landscapes. Traditional fishing landscapes that are attractive to people have intrinsic value. We believe that traditional fishing has an irreplaceable way of being due to its impact on the formation of the local landscape and the beauty of the region.

Based on this consideration, the goal of conserving and passing on the traditional fishery is seeing and researching the traditional fishing fishery as a cultural landscape and maintaining its irreplaceable components. Particularly in terms of culture, diversity of the landscape and sustainability. These are the locations where traditional fishing, such as ICH, can exhibit its special appeal.

Fishing shares characteristics with farming, forestry and animal husbandry that have an impact on the environment and interact with it to produce a culture and landscape that are suitable for that environment. The approach used in this paper is a workable reference when researching the value of safeguarding traditional forestry, agriculture and animal husbandry. On the basis of discussing the economic value, it conducts an in-depth analysis of its various functions, its influence on the formation and maintenance of regional society and the various cultures and landscapes produced.

## 5. Conclusions

By analyzing the practice of beach seine fishing from the perspectives of economic benefit, nature conservation, village formation, the maintenance of social relations and the function of providing various places, it can be seen that traditional fisheries can and do have a multidimensional value.

In terms of the work, culture and landscape of traditional fisheries, the complexity of technology, the creativity and cooperation of natural knowledge, the power of human beings, the diversity of culture and view of life, the impact on the formation of the local landscape and the influence on local beauty are all shown. These are all examples of irreplaceable parts of traditional fishing.

Traditional fisheries are examples of historical culture, cultural landscape and a reflection of a region's charm and individuality. In this study, we have proposed how valuable beach seine fishing has been in the past to coastal communities in Japan and that it is important to recognize and celebrate that value by continuing to support periodic beach seine fishing events. In the future, we propose increasing awareness and research of the related villages, old villas and old shops in the shopping streets to form a broader cultural landscape and more diversified regional resources for tourism development and the enhancement of regional pride.

**Author Contributions:** Conceptualization, L.B.; Methodology, Y.H.; Writing—original draft, L.B.; Supervision, K.I. All authors have read and agreed to the published version of the manuscript.

**Funding:** This research received no external funding.

**Institutional Review Board Statement:** Not applicable.

**Informed Consent Statement:** Not applicable.

**Data Availability Statement:** Not applicable.

**Conflicts of Interest:** The authors declare no conflict of interest.

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
