# Peer review of "The Value of Traditional Fishing from the Perspective of Cultural Heritage: A Case Study of Seine Fishing in Ichinomiya-cho, Kujukuri-hama"

_sustainability, doi:10.3390/su15043855_

Round 1
Reviewer 1 Report
The manuscript represents an interesting analysis of traditional fisheries values from a study site in Japan which is well presented. However, the manuscript misses a well-articulated research question(s) and this means that the study becomes too descriptive. The authors need to clarify the direction of their research (with some pointers provided in point I below); this is not a large change but it needs to be well considered. An opportunity which this manuscript misses is linking this work to the broader cultural fisheries research undertaken many other countries (see point II).
I. Line 99 – 103 – I do not think the research question is clearly defined here. Can you revise this so as to ask a number of questions – it seems to me that the central question is around “irreplaceability” but this is not well articulated in setting out of the study (line 582 – 586, sort of gives you the answer) and you investigate this though looking at economics of traditional fisheries, and additional aspects on the non-economic values. Lines 118 – 127 may also help is articulating the questions.
II. I expect to see in the Discussion a section how you conclusions relate to other work in Japan, and then also globally. There is much work undertaken in other countries – do these studies support your conclusions or does your study come to differing conclusions?? I would suggest two or there paragraphs which put your conclusions in a broader literature analysis here – maybe starting after line 576.
III. Line 160 – 172 – you describe the site but you are not clear why you selected it – is it representative, a standard sandy beach, or suffering from overfishing???? Line 182 – you need to be specific about what information was collected (e.g. fish catches form individual fishermen, market value of catch etc).
Smaller points:
Line 24 – “are earlier” does not make sense
Line 29 – would be good to say that fish protein is much higher, or the main source of protein, in certain communities with suitable reference
Line 30 – better something like “the development of fishing culture has left has also led to a rich cultural heritage..”
Line 38 – not make sense – is Lake Biwa in Nagara Basin?
Line 59 – spell the Convention in full and reference it
Line 62 – refrain from personal indulgence – delete “the most beautiful coastline in the world”
Line 110 – reference this quote
Line 138 – Heidegger needs a reference. Does “irreplaceability” used in the Convention or official documents as well as academic studies?
Line 296 – remove the extra “T”
Line 326 – “not only edible fresh,…”
Line 397 – 457 – this section is too long and can be shortened by 20 – 30%
Reviewer 2 Report
Review of sustainability-2134291
The Value of Traditional Fishing from the Perspective of Cultural Heritage: A Case Study of Seine Fishing in Ichinomiya-cho, Kujukuri-hama
The authors make a well argued case for the heritage value of traditional seine fishing. It is generally well argued.
The introduction must be improved. There is no framing that positions the paper in the existing literature on intangible cultural heritage values in general and the literature on the heritage related to the cultural practices and traditional methods of fishing in particular. Doing so will elevate the paper from a great discussion of Japanese fishing to a major contribution to intangible heritage.
One aspect that should be considered in the framing section on changing perceptions and uses of traditional fishing is that fishing is a livelihood and an outcome, that technologies change and that people adapt. Thus traditional techniques will decline UNLESS they are being valued as a cultural practice in its own right, rather than as a means to an end (ie getting fish for consumption or sale). So what needs to occur, I think, is a brief comment on the nature of
the construction heritage values in general (see https://doi.org/10.1179/2159032X13Z.00000000011 ) and, importantly, that heritage values are mutable qualities that are affected inter-generationally (see https://www.mdpi.com/2571-9408/5/3/105 ), which has specific relevance to the paper under review.
Having read and enjoyed the paper, I felt somewhat ‘let down’ by the Discussion section. Here in particular it is evident that the paper lacks the introductory framing section with existing literature. The discussion should not only address the relevance of the findings in terms of Japanese cultural heritage and its management, but also its relevance to the wider field of tradtional practices and intangible heritage values in general.
As it stands, the authors are doing themselves a great injustice by failing to provide the appropriate framing and discussion. This is great work that can be even better.
Line 30 Why “on the other hand’ this is not contradicting the previous..
Formal issues
The manuscript suffers from some infelicities in expression, and awkward choices of words. I recommend that the paper be edited by a native English speaking professional scientific editor. One aspect a professional scientific editor, rather than someone who looks at mere spelling and choice of words, can do is to tighten the expression and writing. As it stands, some of the paper is verbose and ‘chatty’, as if written for a general interest magazine. The text can be tightened considerably and thus improved.
Round 2
Reviewer 2 Report
sustainability-2134291-peer-review-v2
Second review of
The Value of Traditional Fishing from the Perspective of Cultural Heritage: A Case Study of Seine Fishing in Ichinomi-ya-cho, Kujukuri-hama.
The authors have undertaken a considerable revision of their paper and are to be congratulated on this. However, more needs to be done before the paper can be published.
In my initial review I wrote that
“The introduction must be improved. There is no framing that positions the paper in the existing literature on intangible cultural heritage values in general and the literature on the heritage related to the cultural practices and traditional methods of fishing in particular.”
The authors have undertaken some work on the framing regarding cultural practices and traditional methods of fishing, but the literature review falls well short of what needs to be included. Also, the authors still fail to positions the paper in the existing literature on intangible cultural heritage values in general as well as the mutability of these values (see first my comment in the review), which are central to the paper.
This then carries through to the discussion, there the authors fail to discuss the relevance of their findings not only in terms of Japanese cultural heritage and its management, but also the relevance of their findings to the wider field of traditional practices and intangible heritage values in general.
This will need to be addressed in a second round of revisions
Author Response
对审稿人 2 条评论的回复 2
我们再次感谢您的建设性意见。很抱歉,原稿中的文献综述不清楚。我们已经审查并回应了您提出的要点,并清楚地说明了我们对手稿进行改进的领域:
对评论的回应:
Point 1: The introduction must be improved. There is no framing that positions the paper in the existing literature on intangible cultural heritage values in general and the literature on the heritage related to the cultural practices and traditional methods of fishing in particular.
Point 2: The authors have undertaken some work on the framing regarding cultural practices and traditional methods of fishing, but the literature review falls well short of what needs to be included. Also, the authors still fail to positions the paper in the existing literature on intangible cultural heritage values in general as well as the mutability of these values (see first my comment in the review), which are central to the paper.
Response 1 and 2:
Dear reviewer thank you again for your valuable comments and suggestions. Comments 1 and 2 are responded jointly because they are connected.
As reviewer kindly suggested, we have improved the literature on measuring intangible cultural heritage values and traditional fishing culture. We have added references on the value of traditional fishing as cultural heritage.
We have compiled literature on measuring intangible cultural heritage's value and discovered that most of it focuses on economic value estimation, with only a few studies addressing the evaluation of intrinsic value. As written in the manuscript, studies on the cultural and folkloric values of traditional fishing are scattered in different fields and there are no literature to judge the content related to traditional fishing as a whole. We believe that this is the originality of this manuscript.
The lines referencing the above have been inserted: (in red)
We review the literature on how to measure the value of intangible cultural heritage(ICH). Masoud and Xiao investigated the economic value brought by ICH as cultural capital [12, 13], Heredia-Carroza designs an empirical methodology to measure the perceived value of ICH [14]. Lenzerini noted that to understand the intrinsic meaning and value of ICH requires an understanding of human factors such as self-identification, the history and social evolution of the group corresponding to the heritage, and the close connection of the heritage with the unique identity of its creators and holders [15].
Measuring the value of ICH not only the economic benefits it generates, but also the exploration of its significance to local people and society. This is consistent with the GIAHS and the idea of cultural landscape. A distinctive agricultural system is frequently used to describe traditional cultural landscape [12]. It is a useful starting point for considering this topic from the standpoint of Cultural Landscapes and GIAHS. Cultural landscapes depict a deep and enduring bond between humans and their natural surroundings [5]. Combining GIAHS theories, the 3 primary values of traditional fishing are the most crucial thing to study--(ⅰ) Local and Traditional Knowledge systems(ⅱ) Cultures, Value systems and Social Organizations (ⅲ) Landscapes and Seascapes Feature [4].
When it comes to traditional fishing, related cultures, value systems and social organizations are discussed in the context of folklore, climatology and ecology. The study onAncient Chinese Fisheries illustrated that ancient ecological and ethical knowledge can be found in the conventional fish approach, which has been practiced for millennia [16]. Local fishing communities' resilience to potential climate change is further boosted by the experience that generations of fishermen have amassed [17]. Besides, traditional fishing practices and associated folk activities like sea sacrificial ceremony also strengthen the unity and endeavor among fishermen and generate cohesive force from hearts to build confidence to defeat any terrifying waves and storms [18]. Sidney noted that the traditional fishponds have been serving not only as source of a traditional local food but also adding to the conservation value at large [19]. Mulazzani believed that fishers are often considered to be the guardians of coastal customs, traditions, and of an age-old way of life. Bennett noted that elements linked to cultural heritage may include fishing harbours, boats, fishers themselves, and related buildings, festivals, traditions, traditional trade and barter [20].
These studies on the importance and measure the value of traditional fishing have connections to regional society, folklore, ancient ecological ethics, and ecological conservation. However, no research has examined traditional fishing as a whole; each one of them examines a particular aspect of traditional fishing.
Up to this time, no significant work has been undertaken in the field of assessment of any traditional fishery's total value as intangible cultural heritage and the necessity of preserving it. Besides, none of the above-mentioned approaches takes into consideration from the viewpoint of cultural landscape.
The changes and references in the manuscript can be found on: page 2, lines 97~ page 3 lines 137
In addition, the GIAHS shows that, in a broad sense, forestry, livestock, and fisheries are all a part of agriculture. Studying the historical and cultural significance of agriculture, forestry, and cattle can benefit from using the value of traditional fisheries as a point of reference.
The lines referencing the above have been inserted: (in red)
Additionally, the method for measuring values that was created in this study for traditional fishing might also be used to estimate the value of traditional forestry, agriculture, and animal husbandry as reference.
The changes and references in the manuscript can be found on: page 3 lines 142~144.
Point 3: This then carries through to the discussion; there the authors fail to discuss the relevance of their findings not only in terms of Japanese cultural heritage and its management, but also the relevance of their findings to the wider field of traditional practices and intangible heritage values in general.
Response 3:
Thanks for your comment. According to your suggestion, we have added lines on discussing the relevance of our findings on Japanese cultural heritage and GIAHS.
The lines referencing the above have been inserted:
“Based on this consideration, the goal of conserving and passing on the traditional fishery is seeing and researching the traditional fishing fishery as a cultural landscape and maintaining her irreplaceable components. Particularly in terms of culture, diversity of the landscape, and sustainability. These are the locations where traditional fishing, as ICH, can exhibit its special appeal.
渔业与农业、林业和畜牧业具有相同的特征,它们对环境产生影响并与之相互作用,从而产生适合该环境的文化和景观。本文所采用的方法为研究保护传统林业、农业和畜牧业的价值提供了可操作的参考。在探讨经济价值的基础上,深入分析其各种功能,对区域社会形成和维持的影响,以及所产生的各种文化和景观。”
可以在第 15 页第 646~657 行中找到手稿中的更改和参考。
尊敬的审稿人,再次感谢您提出的宝贵意见和建议。我们已尽力完成所有必需的更改。根据您的建议所做的所有修改已在修订稿中以红色标记。
2023 年 2 月 6 日

Round 3
Reviewer 2 Report
The authors have strived to address the issues I raised in my second review. While I still have reservations, the changes are sufficient to see the paper through to publication
Author Response
Dear Reviewer:
We would like to thank you for you constructive comments and suggestions so far. We have learned much from it.
Thank you once again for your attention to our paper.
Best Regards.
February 11th, 2023.